# A Fine-Grained Image Classification Approach for Dog Feces Using MC-SCMNet under Complex Backgrounds

**DOI:** 10.3390/ani13101660

**Published:** 2023-05-17

**Authors:** Jinyu Liang, Weiwei Cai, Zhuonong Xu, Guoxiong Zhou, Johnny Li, Zuofu Xiang

**Affiliations:** 1College of Computer & Information Engineering, Central South University of Forestry and Technology, Changsha 410004, China; 2School of Artificial Intelligence and Computer Science, Jiangnan University, Wuxi 214122, China; 3Department of Soil and Water Systems, University of Idaho, Moscow, ID 83844, USA; 4Institute of Evolutionary Ecology and Conservation Biology, Central South University of Forestry and Technology, Changsha 410004, China

**Keywords:** dogs, gastrointestinal monitoring, fine-grained image classification, complex backgrounds, fecal identification, deep learning

## Abstract

**Simple Summary:**

Currently, more and more people keep dogs, and the gastrointestinal diseases of pet dogs have brought great losses to families. However, the condition of the dog’s feces is closely related to the health of its stomach and intestines. We can know the intestinal condition of dogs in advance by scoring dog feces, and implement measures such as food adjustments. The PURINA FECAL SCORING CHART and the WALTHAM™ Faeces Scoring System are good at scoring dog feces visually, but some scoring experience is required. Therefore, this paper proposes an artificial intelligence method to automatically classify the condition of dog feces by combining their classification criteria with the assistance of animal experts. This method can achieve an accuracy of 88.27%, improving the diagnostic efficiency of veterinarians.

**Abstract:**

In a natural environment, factors such as weathering and sun exposure will degrade the characteristics of dog feces; disturbances such as decaying wood and dirt are likely to make false detections; the recognition distinctions between different kinds of feces are slight. To address these issues, this paper proposes a fine-grained image classification approach for dog feces using MC-SCMNet under complex backgrounds. First, a multi-scale attention down-sampling module (MADM) is proposed. It carefully retrieves tiny feces feature information. Second, a coordinate location attention mechanism (CLAM) is proposed. It inhibits the entry of disturbance information into the network’s feature layer. Then, an SCM-Block containing MADM and CLAM is proposed. We utilized the block to construct a new backbone network to increase the efficiency of fecal feature fusion in dogs. Throughout the network, we decrease the number of parameters using depthwise separable convolution (DSC). In conclusion, MC-SCMNet outperforms all other models in terms of accuracy. On our self-built DFML dataset, it achieves an average identification accuracy of 88.27% and an $F_{1}$ value of 88.91%. The results of the experiments demonstrate that it is more appropriate for dog fecal identification and maintains stable results even in complex backgrounds, which may be applied to dog gastrointestinal health checks.

## 1. Introduction

Dogs are one of the first domesticated animals in the world and are widely bred for their docile temperament. Living with a pet dog can relieve stress [1] and even provide medical psychological support [2]. Therefore, people pay more and more attention to the health and physical condition of dogs [3,4]. In the breeding process, if the dog is sick and not treated in time, the breeder will suffer both economic and spiritual losses [5]. Therefore, it is very important to discover the dog’s condition in time and take treatment measures in advance. Due to the complex and changeable breeding environment or the different physical levels of dogs, dogs are often affected by parasitic infections [6], gastrointestinal biota disorders (dysbiosis) [7], stress disorders [8], etc., often accompanied by various gastrointestinal diseases [9], such as irritable bowel syndrome [10], chronic enteropathy [11], chronic diarrhea [12], etc. Observing the shape and consistency of their feces can reveal the characteristics of gut microbes [13], which can help breeders and veterinarians judge the dog’s intestinal status [14], and provide feedback on the abnormal health of the dog’s gastrointestinal tract.

At present, the analysis methods of feces include: experimental analysis, manual detection, machine learning, and deep learning. Experimental analysis can comprehensively detect the microorganisms and chemical components in feces [15], but it can only rely on professional equipment and medical laboratories [16], and the detection cost is high. The vast majority of methods are manual inspection, which is usually visually judged by a veterinarian on the shape, size, quality, and color of a fecal sample [17]. Based on this information, veterinarians can give early warning of potential crises in dogs’ intestinal health and provide dogs with targeted food menus [18]. The PURINA FECAL SCORING CHART [19] and the WALTHAM™ Faeces Scoring System [20] are the most common stool scoring manuals. Stool score can reflect intestinal health [21]. According to them, novices can also classify dog feces, but certain experience in judging is also required, because the manual only provides individual pictures for reference [22]. However, manual stool analysis still requires a significant amount of time, even when scored according to the manual. Analyzing feces day in and day out is embarrassing, unsanitary, labor-intensive, and difficult for the owner to maintain [23,24]. In addition, because of the inconsistency of classification standards, human subjective judgment will also affect the accuracy of classification [22]. With the development of computer technology, many researchers have introduced machine learning methods into the field of healthcare [25]. This makes it possible to automatically classify images of dog feces. Humans automatically classify input feces images by manually designing features. This approach greatly speeds up the scoring of stools. However, handcrafted features are complex and have limited detection accuracy. The deep learning that emerged later, its end-to-end network structure simplifies manual feature design and further improves the accuracy and speed of classification. This paper will provide a reference for the automatic classification task of dog feces by image classification technology based on deep learning. The method in this paper can provide a priori screening before a comprehensive diagnosis by a veterinarian, and achieve an initial early warning of gastrointestinal diseases in dogs [26]. In the absence of a veterinarian, or for an owner without a medical background, the method in this paper can also be used to evaluate the dog’s feces and make targeted food adjustments [27].

The present FGVC (Fine-Grained Visual Categorization) of dog feces (Figure 1) has three problems that need to be addressed: (1) Natural environment can cause qualitative changes in feces. Dog feces are usually seen in outdoor areas. Freshly ejected feces have distinct characteristics; however, when exposed to natural settings such as wind drying, sunshine, and rain, the available information for identification is loss. This makes it difficult to gather enough information for standard network down-sampling operations [28,29]. (2) Misidentification of things as dog excrement. The environment in which feces are located is complex and diverse. In terms of morphological trait (shape, color, etc.), dead leaves, dirt, and broken limbs are similar to dog excrement. These items can be particularly disruptive to the categorization process when they exist simultaneously [30]. (3) Less interclass variation in dog feces. When a dog is moderately unwell, its feces are comparable to healthy ones. Only a very experienced veterinarian can correctly anticipate a dog’s illness at this time [31]. This can make checking a dog’s health condition considerably more costly.

To address the issue that the features of degraded dog feces are hard to retain via down-sampling, we propose a MADM method that combines the advantages of convolution and pooling operations. The method parallelizes multiple convolution and maximum pooling while concatenating an attention mechanism. Such a connected structure collects global features and enhances the scale variety of local features. As a result, the down-sampled feces information is fully represented, and critical information is highlighted. Despite being transformed by harsh natural environments such as air-drying, MADM can keep its fine traits to the best of its ability.

To solve the issue of false identification of dog feces, we propose a novel attention mechanism CLAM based on the properties of dog feces that filters background interference and improves the network model’s awareness to fecal properties. We propose a novel attention mechanism CLAM based on the properties of dog feces that filters background interference and overcomes the problem of false identification of dog feces. Because feces are separated from the ground at different heights, their spatial distribution differs. Moreover, CLAM has a cross structure that can swiftly perceive spatial information and provides auxiliary network features. Based on CA’s location weight assignment strategy [32], CLAM concentrates spatial attention by embedding global and local information into feature channels along horizontal and vertical directions, respectively. This helps the network to focus on the given objects while minimizing the interference of other redundant inputs.

To overcome the inter-class similarity problem of dog feces, we propose a brand new SCM Block. Specifically, we combine the previous CLAM and MADM to construct a new feature extraction unit, recreate the ResNeSt backbone network, and compose SCMNet. Such a design is end-to-end without additional computational overhead for complex backgrounds in images. SCMNet improves the network’s ability to express stool information by aggregating CLAM attention weights and MADM multi-scale features. It boosts the accuracy of ResNeSt’s adaptation to dog feces.

The research contributions of this paper can be summarized as follows:
For the first time, we collected a dataset of 1623 images of dog feces (DFML). According to the suggestion of Zuofu Xiang [33,34], a member of the Institute of Evolutionary Ecology and Conservation Biology, Central South University of Forestry and Technology, the dataset divides dog feces into four categories: diarrhea, lack of water, normal, and soft stool, providing a reference for the gastrointestinal diagnosis of dogs.In order to address the challenges of dog feces image identification, we suggest the MC-SCMNet network, which is designed as follows:
We design a multi-scale attention down-sampling module (MADM) to address the problem of losing critical characteristics owing to the degradation of dog feces. It enables the network to keep fecal texture details to the maximum extent possible while also fighting the impacts of fecal feature degradation.We design a coordinate location attention mechanism (CLAM) to minimize the interference of similar items on dog feces identification. Its unique cross structure is able to filter noisy signals while highlighting fecal feature values in the attention map.We design a SCM-Block to establish a new backbone network SCMNet to address the problem of classification mistakes caused by slight variations across dog feces categories. This block combines CLAM’s spatial attention information and MADM’s multi-scale features, improving the fusion efficiency of dog feces characteristics.We employ depthwise separable convolution (DSC) to compensate for the increase in parameter number and minimize network training time and hardware costs while improving MC-SCMNet flexibility.This approach achieved an average recognition accuracy of 88.27% and an $F_{1}$ score of 88.91% on DFML data. In comparison to other approaches, MC-SCMNet showed the best accuracy and pertinence in the categorization of dog feces in natural contexts. This serves as a prototype for the application of deep learning technology to the categorization of dog feces, helping farmers to swiftly check the intestinal health of their dogs.

## 2. Related Works

The problem of classifying dog feces is a typical fine-grained visual classification (FGVC) problem [35]. FGVC focuses on samples of the same or closely related subordinate categories and is more specific than standard image classification hierarchies. For example, classifying different vehicle models [36], dog breeds [37], fruit types [38], citrus surface defects [39], etc. Fine-grained image classification has undergone long-term development, transitioning from machine learning methods to deep learning. 

In recent years, researchers have applied complex machine learning [40,41,42,43,44,45,46,47,48,49] to the field of fine-grained image classification, which provides a reference for our research. In terms of agriculture, Chen et al. [28] used a tomato leaf disease recognition method based on the combination of ABCKBWTR and B-ARNet, and achieved a recognition accuracy of about 89%. Li et al. [50] proposed a FWDGAN method combining the deep features of ResNet and the global features of InceptionV1. This method achieved high recognition accuracy in tomato disease recognition. Deng et al. [51] integrated DenseNet-121, SE-ResNet-50, and ResNeSt-50 to diagnose six types of rice diseases, with an average accuracy of 91%. In the related field of stool, Saboo et al. [52] used machine learning (ML) analysis to find that the fecal microbiota was strongly correlated with the severity of liver cirrhosis. Ludwig et al. [53] used machine learning to classify infant feces, achieving 77.0% of manual detection. Hwang et al. [54] used a support vector machine (SVM) classifier to classify feces in colonoscopy videos based on color features, and the performance average accuracy of Sensitivity and Specificity reached 88.89 and 91.35, respectively. Liao et al. [55] developed a stool diagnostic system, which can achieve 100% and 99.2% recognition accuracy in color and features of stool images. Zhou et al. [56] proposed a hierarchical convolutional neural network architecture to train stool shapes. Their method also incorporated machine learning, and finally achieved 84.4% accuracy and 84.2% latency reduction on human stool classification. Leng et al. [57] achieved a classification accuracy of 98.4% on a self-built dataset of five stool classifications through a lightweight shallow CNN. Choy et al. [58] used ResNext-50 to classify human feces for monitoring and diagnosis of human diseases, achieving an accuracy of 94.35%.

However, the previous investigations are primarily focused on a single background interference, and the classification subject is merely human excrement, which is not extensible. Furthermore, applying solely deep learning components does not explore the relationship between stool features and network structure, and it is incapable of coping with more complex background interference. Therefore, this paper extends the target to dog feces, and selects ResNeSt as the benchmark network to broaden the application range of FGVC.

## 3. Materials and Methods

### 3.1. Materials Acquisition

The dataset of dog feces served as the foundation for the paper. We collected a significant number of dog feces photos to create the dog feces differentiated by moisture level (DFML) dataset. We based on the PURINA FECAL SCORING CHART [19] and the WALTHAM™ Faeces Scoring System [20]. On the advice of Zuofu Xiang [59,60], the stool images were classified into four categories based on their moisture contents: diarrhea, lack of water, normal, and soft stools. In addition, we cleaned invalid images according to the following criteria: (1) complete loss of stool characteristics; (2) images that are wobbly and blurry; (3) excessive light exposure or inadequate light. Finally, 1623 valid photos were obtained, and Table 1 shows the features and amounts of images in each category. The images in the DFML dataset consist of two parts. One part was captured by Canon EOS R6 in natural light, with 1412 images from Hunan Provincial Animal Hospital, White Sand Kennel, Changsha Small Animal Protection Association; the other part was screened on the Internet, with 211 images.

The following are detailed descriptions of the four categories of feces: (1) Lack of water, with a dry surface and a hard texture. Sometimes the feces are roughly spherical in form, while other times it resembles a concave-convex strip. (2) Normal dog feces have visible gaps, commonly in the shape of a long snake. (3) Soft stools have a smooth and soft texture, a shiny finish, and an irregular form. (4) Diarrhea is sticky, extremely soft and shapeless, and occasionally liquid. There are also similarities in the characteristics of the categories, such as (1) and (2) lengthy strips, (2) and (3) similar surface color, and (3) and (4) shapelessness. These traits cause disturbances in the categorization of dog feces and researching them has practical significance for FGVC.

### 3.2. Materials Preprocessing

A large number of images can improve the network’s training process by enriching the effective characteristics of feces and reducing the problem of overfitting. For data augmentation of DFML images, we utilize the imgaug package and use the following methods at random: (1) 50% Gaussian blur with coefficients δ∈[0,0.5]; (2) Enhance or reduce the image’s contrast; (3) Gaussian noise, 50% of the images sample the noise once every pixel, other images sample the noise every pixel and channel; (4) Adjusts the brightness of 20% for the images; (5) Affine transformations, including scaling, panning, rotating, and clipping. Finally, we gathered 3246 training images, and Figure 2 and Table 2 illustrate the enhancing effect and augmented data.

### 3.3. MC-SCMNet

This paper proposes a network MC-SCMNet based on ResNeSt for FGVC of dog. Figure 3 depicts the model structure, with (a) to (e) representing the main structure, MADM module, CLAM, DSC, and SCM-Block. In the feature extraction stage, we utilize MADM and CLAM to enrich feces features while filtering out irrelevant information to enter the feature fusion stage. We employ SCM-Block to build a backbone network during the feature fusion stage. SCM-Block combines the advantages of Split-Attention, CLAM, and MADM, optimizes fusion efficiency, accelerates information sharing across layers, and improves the network’s global perception ability. Furthermore, DSC was used to reduce the number of network parameters and expand the flexibility of MC-SCMNet. MC-SCMNet integrates dog feces characteristics more effectively than ResNeSt to strengthen adaptive judgment for the future classification stage.

#### 3.3.1. Multi-Scale Attention Down-Sampling (MADM)

The natural environment’s attenuation of feces characteristics has a stronger impact on the network feature. The down-sampling structure in the neural network can adjust the image feature scale, create picture thumbnails, and extract prominent features in the receptive field. It can highlight valuable information and help to ease this condition. Scaling the picture size can reduce complexity and suppress overfitting. Down-sampling is commonly done with maximum or average pooling. The former emphasizes the most prominent characteristics in the receptive field, while minor feature information is readily overlooked. The latter is milder, smoothing all information and diminishing the sharpness of prominent characteristics. However, this will impair the network’s translation invariance, making it difficult to withstand noise interference. Specially, weathering, rain, and sun, for example, will dilute the texture and color features of feces. This lowers the network’s effective feature values and makes judging tougher for the classifier. Direct traditional down-sampling of feces images will exacerbate the loss of tiny features at this stage. Furthermore, standard pooling is static and does not have learnable parameters. In summary, the natural attenuation of stool characteristics reduces feature information, and standard down-sampling is difficult to cope with. As a result, we propose a MADM and the structure of which is shown in Figure 3b.

We divide the information flow into three branches, with a different kernel of convolution and maximum pooling. Unlike the static pooling process, the convolution operation can share all of the sampled points in the feature map, boosting the model’s attention on tiny fecal characteristics. Furthermore, multiple convolutions can extract multi-scale information and comprehensively describe fecal characteristics. We employ large convolution kernels of 5×5 and 7×7 with a stride of two. Large convolution, on the other hand, increases the amount of network parameters, which we balance by employing depth-separable convolution. These two branches are trainable and compensate for the static disadvantage of max-pooling. This enables MADM to extract the most important feces textures while merging rich dog feces features to add data to the network. Finally, we employ SE attention to enhance the correlation between fecal feature channels and the information flow between the three branches.

#### 3.3.2. Coordinate Location Attention Mechanism (CLAM)

In the environment, dead leaves, dirt, stumps, and other similar items often hamper dog feces categorization. We can employ attention mechanisms in deep learning to filter out these disturbances. The Attention Mechanism (AM) may reconfigure the corresponding weight of the feature map based on the target’s attributes, therefore strengthening the target’s feature expression. CA is a fairly new attention mechanism. It decomposes channel attention into two intersecting one-dimensional codes, aggregates characteristics in two spatial directions, and can locate target information accurately. Its spatial sensitivity is beneficial for fine-grained categorization in complicated environments. We propose a CLAM based on CA and the background properties of dog feces, and the structure is depicted in Figure 3c.

The feature map with input size h×w×C is first duplicated three times. The coordinate information $x_{C}$ is initially embedded in the feature map, as illustrated in Equation (1):(1)$$Z_{C}=\frac{1}{h\times w}\sum_{i=1}^{h}\sum_{j=1}^{w}x_{C}\left(i,j\right)$$

Then, average pooling is employed to provide direction-awareness for both copies along the horizontal and vertical dimensions. The orientation perception is made up of two one-dimensional feature encodings, one in each direction. The feature encoding output in horizontal coordinate h and vertical coordinate w is as follows:(2)$$Z_{C}^{h}\left(h\right)=\frac{1}{w}\sum_{j=1}^{w}x_{C}\left(h,j\right)$$
(3)$$Z_{C}^{w}\left(w\right)=\frac{1}{h}\sum_{i=1}^{h}x_{C}\left(i,w\right)$$

We transpose the horizontal coordinate codes to allow later dimensional unity. Equation (4) depicts the transposition process.
(4)$${\left[Z_{C}^{h}\left(h\right)\right]}^{T}=\frac{1}{w}\sum_{j=1}^{w}x_{C}\left(j,h\right)$$

We design a location attention mechanism (LAM) as the core of CLAM. This mechanism contains DSC and dynamic connection to generate changeable dimensionality via global pooling and parameter size. It enriches the fecal information adaptively. The pooling result is then coupled to a fully connected layer and encoded as a one-dimensional feature. Simultaneously, we join a 3×3 DSC with a dot product one-dimensional encoding. Following that, we synthesize the output LAM horizontal and vertical direction information, and the fusion process is depicted in Equations (5) and (6).
(5)$$A_{A}=M\left\{P_{a}\left[P_{1}\left(F_{1}\right)\right],C_{1}\left[P_{1}\left(F_{1}\right)\right]\right\}$$
(6)$$A_{B}=M\left\{P_{b}\left[P_{2}\left(F_{2}\right)\right],C_{2}\left[P_{2}\left(F_{2}\right)\right]\right\}$$
where $F_{1}$ and $F_{2}$ are LAM inputs, $P_{1}$ and $P_{2}$ are average pooling, $P_{a}$ and $P_{b}$ are fully connected layers, M is the element dot product, and $A_{A}$ and $A_{B}$ are LAM output information.

Finally, we utilize the multiplication operation and softmax to fuse the LAM information in two spatial directions and aggregate the output $A_{C}$ of the third branch to generate the CLAM module’s output $F_{C}$, as indicated in Equation (7).
(7)$$F_{C}=\left\{softmax\left[M{(A}_{A},A_{B}\right)\right],A_{C}\}$$

CLAM module output comprises multiple spatial angles of feces and significant target location, efficiently filtering the interference of complex backdrop and solving the problem of mistaken identification of identical items.

#### 3.3.3. SCM-Block

The color, posture, and texture characteristics of dog feces are quite complex, with slight variances across classes as well as diversity within classes. For example, water-deficient and normal feces are generally long strips; diarrhea is yellow, black, brown, and green in hue. Their color map spaces and texture channels are extremely comparable for the similarity problem. If the feature fusion is insufficient, the model will ignore minor variations, affecting the model’s performance.

Therefore, we propose a new SCM-Block based on ResNeSt-Block, which stacked a new four-layer backbone network SCMNet. The block is based on Split-Attention for channel information modeling, combining the spatial dimension of CLAM and the multiscale information of MADM to improve information exchange efficiency, and its structure is shown in Figure 3e. CLAM first locates the fecal target in the input image, and MADM then stretches the feces’ feature scale. Finally, we transmit the combined information into the Split-Attention module. For the input map X, the procedure is calculated as follows.

In the residual branch, different stride settings execute distinct procedures. With Stride=1, the network maps the information directly; when Stride=2, the network down-samples using MADM, as indicated in Equation (8).
(8)$$T\left(X\right)=\left\{\begin{array}{ll} X & if\;Stride=1,\\ MADS\left(X\right) & if\;Stride=2, \end{array}\right.$$

In the main branch, we first use CLAM to extract the position information from the input image, and then use 1×1 convolution to split the feature channel into k×r branches (hyperparameters Cardinal=k, Radix=r). When the stride is 2, we utilize MADM to fuse the down-sampled features; when the stride is 1, we use DSC with kernel 3×3 to fuse the features. The final k×r sets of multi-channel characteristics are transmitted to the Split-Attention module. The Split-Attention calculating method is as follows:The Split Attention of each group of Cardinals is added to each split element, and the k group of Cardinal Groups is shown in Equation (9).


(9)$${\hat{U}}^{k}=\sum_{j=R\left(k-1\right)+1}^{Rk}U_{j}$$
where, ${\hat{U}}^{k}\epsilon R^{H\times W\times C/K}\;for\;k\epsilon 1,\;2,\;\ldots K$, H, W and C is the size of the block output feature map.


2.Global information is gathered by pooling the global average of $s^{k}\epsilon R^{C/K}$ across spatial dimensions, with the cth component derived as defined in Equation (10).




(10)
$$s_{c}^{k}=\frac{1}{H\times W}\sum_{i=1}^{H}\sum_{j=1}^{W}{\hat{U}}_{c}^{k}\left(i,j\right)$$




3.The channel is separated into r branches using an 1×1 convolution kernel, and the weights are allocated by the R-Softmax operation after 1×1 convolution, as described in Equation (11). $G_{i}^{c}$ ic indicates calculating the weight of each split for $s^{k}$.




(11)
$$a_{i}^{k}\left(c\right)=\left\{\begin{array}{cc} \frac{\exp\left(G_{i}^{c}\left(s^{k}\right)\right)}{\sum_{j=0}^{r}\exp\left(G_{j}^{c}\left(s^{k}\right)\right)} & if\;r>1,\\ \frac{1}{1+\exp\left(-G_{i}^{c}\left(s^{k}\right)\right)} & if\;r=1, \end{array}\right.$$




4.We sum the characteristics to get the cth channel calculation formula, which is shown in Equation (12).




(12)
$$V_{c}^{k}=\sum_{i=1}^{r}{a_{i}^{k}\left(c\right)U}_{R\left(k-1\right)+i}$$



The Split-Attention output of each group is then aggregated along the channel dimension as illustrated in Equation (13), and the backbone network output information is acquired after passing through the 1×1 convolution kernel.
(13)$$V=Concat\left\{V^{1},V^{2},\ldots V^{k}\right\}$$

We conduct feature fusion on the residual and main branches, and the final output signal Y is presented in Equation (14).
(14)$$Y=T\left(X\right)+V$$

The number of split channels is determined by the SCM-Block hyperparameters cardinal and radix pair, which has different effects on model performance. The discussion of hyperparameters can be found in Section 4.3.

#### 3.3.4. Depthwise Separable Convolution (DSC)

MC-SCMNet increases the amount of network parameters while boosting accuracy, which makes fast training difficult. As a result, we use DSC instead of standard convolution (as shown in Figure 3d) to decrease the number of model parameters and optimize the training process without impacting performance. DSC consists of a sequential connection between depthwise convolution and pointwise convolution. The former is a spatial convolution, with each input channel being individually conducted. The latter is an 1×1 convolution that reflects the input channels of the previous operation into a new channel space. It is similar to a 3D filter in that it has both channel and spatial dimensions. DSC introduces grouped convolution to the standard convolution. This allows it to calculate feature mapping and spatial mapping independently. As a consequence, it has no effect on the outcome of information processing and reduces the number of parameters.

Consider the input data to be S×S×C, the convolution kernel to be K×K×N, the stride to be 1, and the Equation (15) for the number of standard convolution parameters.
(15)$$W_{SC}=K\times K\times C\times N$$

The calculation volume corresponding to the standard convolution is shown in Equation (16).
(16)$$O_{SC}=S\times S\times K\times K\times C\times N$$

The parameter number of DSC is shown in Equation (17).
(17)$$W_{DSC}=K\times K\times C+C\times N$$

The corresponding calculation amount of DSC is shown in Equation (18).
(18)$$O_{DSC}=S\times S\times K\times K\times C+S\times S\times C\times N$$

Therefore, the ratio between the number of parameters and the amount of computation corresponding to the two structures is shown in Equations (19) and (20).
(19)$$F_{W}=\frac{W_{DSC}}{W_{SC}}=\frac{1}{N}+\frac{1}{K^{2}}$$
(20)$$F_{O}=\frac{O_{DSC}}{O_{SC}}=\frac{1}{N}+\frac{1}{K^{2}}$$

The above formulas show that DSC converts multiplication in convolution to addition, obtaining the same result. It makes better use of model parameters, lowers operational costs, and accelerates computation speed.

## 4. Results and Analysis

This section is divided into ten subsections: (1) Experimental environment and preparation; (2) Evaluation indicators; (3) Hyperparameter selection; (4) Data validation methods and tests; (5) Ablation experiment; (6) Analysis of performance experiments; (7) Comparison with other state-of-the-art methods; (8) Statistical test; (9) Model visualization; and (10) Workflow for a practical application. We tested and compared the models in many aspects. The results show that MC-SCMNet handles the problems of qualitative changes in dog feces in the natural environment, misdetection of objects similar to it, and tiny variability between classes.

### 4.1. Experimental Environment and Preparation

To prevent different experimental conditions from influencing MC-SCMNet results, all experiments in this paper are done in the same hardware and software environment. The main hardware devices used in this experiment are NVIDIA GeForce RTX 3070 Ti and 12th Gen Intel(R) Core (TM) i9-12900H 2.50 GHz CPU (Legion Y9000P|AH7H). The main software devices need to be compatible with the specific hardware. Table 3 depicts the particular experimental environment of this work. Considering the hardware performance and training effect, this paper sets the experimental batch to 130, employs the Adam optimizer, the learning rate is 0.0001, and the batch size is set to 8.

### 4.2. Evaluation Indicators

In this paper, we evaluate the performance of the model by accuracy, precision, recall, $F_{1}$, parameter size, and training time.
(21)$$Accuracy=\frac{TP+TN}{TP+TN+FP+FN}$$
(22)$$Precision=\frac{TP}{TP+FP}$$
(23)$$Recall=\frac{TP}{TP+FN}$$
(24)$$F_{1}=2\times \frac{Precision\times Recall}{Precision+Recall}=\frac{2\times TP}{2\times TP+FP+FN}$$
where TP is true positive: prediction is positive, actuality is also positive; FP is false positive: predicted positive, actual negative; TN is true negative: predicted negative, actually negative; FN is false negative: predicted negative, actual positive. The inter-class average of model accuracy and recall is calculated using the weighted-average strategy [61]. The parameter is used to measure the model size and operating cost. Training time indicates the model’s training speed.

### 4.3. Hyperparameter Selection

In Section 3.3.3, we mentioned the different effects of hyperparameters on the model. In order to explore the optimal performance combination, we test the impacts of various radix and cardinal models, and the experimental results are listed in Table 4.

According to the experimental results, the network performs best when radix = 2 and cardinal = 1. Therefore, the combination of radix = 2 and cardinal = 1 was used for all subsequent experiments in this paper.

### 4.4. Data Validation Methods and Tests

We put the data-augmented images, a total of 3246 dog fecal photos in four categories, into the model training while keeping the other experimental settings constant. The picture size has been standardized at 224 × 224. The 5-fold cross-validation approach is utilized for training in this paper. First, the images are split into five equal pieces at random. Then, five repeat trials were set up to train four pieces in sequence, leaving one to test the model. The model’s ultimate performance is measured using the average of five experimental outcomes. This strategy efficiently avoids the contingency of model training while improving experiment accuracy and dependability.

We utilize this strategy to train ResNet50, ResNeSt50, and MC-SCMNet, as shown in Figure 4, and their average accuracy is 80.22%, 83.70%, and 88.24%, respectively. MC-SCMNet’s accuracy is much higher than that of the original models ResNet50 and ResNeSt50. To further validate the performance of our model, we retrained ResNet50, ResNeSt50, and MC-SCMNet using the images before data augmentation; the resulting accuracy was 71.30%, 74.69%, and 77.30%, respectively. Experiments demonstrate that our model outperforms ResNet50 (+5.86%, +8.02%) and ResNeSt50 (+2.47%, +4.54%) on both the original and augmented datasets. Furthermore, as the number of data images increases, so does the improvement in accuracy.

The MC-SCMNet model can achieve an average accuracy of 88.24% after training with five-fold cross-validation. Fold 1’s accuracy is 88.27%, which is the closest to the average accuracy. Figure 5 depicts the training procedure. The accuracy and loss of the validation set and training set tend to converge as the number of training iterations increases in the figure. MC-SCMNet’s training accuracy reaches 97.11% after 130 epochs.

### 4.5. Ablation Experiment

We present MADM and CLAM modules, build SCM-Block, and apply DSC to the models in this paper. To test their effectiveness, we chose four metrics to conduct ablation experiments on the model: Block, Attention, Conv, and under-sampling. Table 5 displays the experimental outcomes.

Table 5 shows a total of 14 experimental schemes. Schemes 1 and 2 are the basic models ResNet50 and the baseline ResNeSt50, respectively. Schemes 8 and 14 are the intermediary models SCMNet and MC-SCMNet, respectively.

When comparing Scheme 4 and Schemes 1–3, it is shown that Split-Attention + CLAM can execute feature extraction better, with a considerably higher accuracy than no attention (+5.55%), Split-Attention (+2.16%), and Split-Attention + CA (+1.54%). When comparing Schemes 5–7 and Schemes 2–4, it is discovered that DSC significantly decreases the amount of network parameters (−8.861 M) while maintaining same identification accuracy. When Schemes 2 and 8 are compared, it is shown that SCM-Block can better filter the interference information in the feature fusion stage of the model, decrease feature information loss, and greatly enhance model accuracy (+3.70%). When comparing Schemes 8 and 9, we discovered that employing MADM allows us to better focus on the model’s tiny information during the down-sampling phase. Schemes 8–14 completely depict the effect of MADM, CLAM, and DSC on SCM-Block on the model. In Scheme 14, MC-SCMNet has an accuracy of 88.27% and greatly surpasses ResNet50 (+8.02%) and ResNeSt50 (+4.63%).

### 4.6. Analysis of Performance Experiments

#### 4.6.1. Comparison with Other Basic Classification Networks

In terms of average accuracy, we compare MC-SCMNet with SCMNet and some basic networks: AlexNet, VGG, Googlenet, ResNet, and ResneSt. To clearly depict the curve values, we apply the same level of high-dimensional smoothing and fitting to them. After processing, the average accuracy of the MC-SCMNet and SCMNet networks, as well as the AlexNet, VGG, Googlenet, ResNet, and ResneSt networks, are depicted in Figure 6 by red, blue, orange, purple, green, brown, and black dotted lines. We can clearly observe that the accuracy of the SCMNet network model is substantially greater than that of the ResNeSt model after 130 epochs. This demonstrates that SCM-Block has a considerable effectiveness in properly identifying stool types.

Table 6 displays their accuracies on different types of data. We constructed the MC-SCMNet model, which has the maximum accuracy in diarrhea (93.65%), normal (85.71%), and soft stool (84.71%). However, the model’s accuracy on water-deficient stool (92.06%) is somewhat lower than that of SCMNet (93.65%) and ResNeSt-50 (93.12%). This is because water-deficient and normal stool samples have similar features, and the number is much higher than that of diarrhea and soft stools. ResNeSt and SCMNet’s data extraction stage (deep-stem) pays insufficient attention to location information and down-sampling details. This enables the identification of a large number of normal-type feces with similar characteristics as dry. As a result, the water-deficient type’s accuracy is artificially high, while the normal type’s accuracy is low.

#### 4.6.2. The Model’s Evaluation Parameters and the Confusion Matrix

As indicated in Table 7, we also compared the MC-SCMNet model under each classification to the initial model ResNet50 and the basic model ResNeSt50. Figure 7 depicts the confusion matrix of different categories of the MC-SCMNet model and the original models.

The accuracy of ResNet50 ranges from 71% to 88%, with an average accuracy of 80.79%; the recall ranges from 71% to 89%, with an average recall of 81.16%. The model’s $F_{1}$ value is 80.98%, and its accuracy is 80.25%. The accuracy of ResNeSt50 ranges from 76% to 93%, with an average accuracy of 84.13%; the recall ranges from 76% to 90%, with an average recall of 84.05%. The model’s $F_{1}$ value is 84.09%, and its accuracy is 83.64%.

MC-SCMNet, on the other hand, maintains a precision rate of 84% to 94%, with an average precision rate of 88.89%; and a recall rate of 85% to 95%, with an average recall rate of 88.93%. The model’s $F_{1}$ value is 88.91%, and its accuracy rate is 88.27%. The testing results show that MC-SCMNet can distinguish the feces features of each category accurately and increases the lower bound of accuracy and recall. The metrics shown above are much higher than ResNet50 and ResNeSt50. It thoroughly demonstrates that MC-SCMNet is effective.

### 4.7. Comparison with Other State-of-the-Art Methods

In order to better assess MC-SCMNet’s classification ability in dog feces. It is compared to ResNeXt50 [48], B-ARNet [28], DMS-Robust Alexnet [62], Swin-transformer [63], and CA-MSNet [64]. Table 8 shows that the recognition accuracy of MC-SCMNet in each category is higher than that of other state-of-the-art methods. The network’s SCM-Block combines CLAM for spatial information and MADM for small information utilizing the Split-Attention approach. This raises the computational cost while also improving network performance, resulting in a long training time (3 h 31 min 38 s). As a result, in the future, we will employ more lightweight strategies to improve the network’s categorization performance and training speed.

### 4.8. Statistical Test

In order to rule out the randomness and chance of our experimental data, we use SPSS to conduct One-way ANOVA of variance on the experimental data in Table 6 and Table 8 [65], the experimental results are shown in Table 9. We first made a null hypothesis ($H_{0}$) that all performance indicators were equal and any small gains or losses observed were not statistically significant, and made a valid hypothesis ($H_{1}$) that the observed gains or losses are statistically significant. Finally, set ∝=0.05. The results show: F=4.1364 > Fcrit=2.0666, p=0.0000579<∝=0.05, so reject $H_{0}$. Therefore, we believe that there are significant differences between these groups, and that MC-SCMNet is superior to the current common mainstream methods, and is pertinent to feces.

### 4.9. Model Visualization

In order to more intuitively observe the concentration of the MC-SCMNet model, we produced a heat map using the Gradient-based Classification Activation (Grad-CAM) method and assessed the model in conjunction with the original image. Figure 8 shows the extracted feature layers of the MC-SCMNet model, the original model ResNet50, and the basic model ResNeSt50 for comparison.

We can observe in the MC-SCMNet model that after layer 1 training, the network’s attention is somewhat divergent, focused on detailed information. Following layer 2 training, the network pays more attention to the features of the darker portions while also paying attention to the model’s details. The network begins to focus on the edge boundaries of objects in the picture after layer 3 training. Following layer 4 training, the network completes picture feature extraction and concentrates all attention on the dog feces region. After layer 1 training, the attention of the ResNet50 and ResNeSt50 models is concentrated on a narrow local area, and the entire attention to the image is lost. After layer 2 training, ResNet50 concentrates on observing weeds and water stains; the ResNeSt50 model’s attention begins to diverge, focusing more on water stains and excrement. Following layer 3 and layer 4 training, the two models concentrate on detecting ground water spots.

ResNet50 and ResNeSt50 highlight erroneous interference information based on the findings stated above. During the model training process, MC-SCMNet pays close attention to the characteristics of location information and down-sampling details, which filters out the influence of similar items and highlights the properties of feces. As a result, MC-SCMNet is more adapted to dog feces identification than ResNeSt50 and ResNet50.

To examine the reliability of MC-SCMNet in dog feces categorization further, we visually compared it to AlexNet, VGG-16, GoogleNet, ResNet-50, ResNeXt-50, and ResNeSt-50 models. Figure 9 depicts the experimental outcomes. We discovered, via observation, that the distractors in the image contributed to the incorrect attention of other categorization algorithms to some extent. MC-SCMNet, on the other hand, may totally suppress the expression of background information and specifically focus on the feces section, boosting feces classification accuracy.

### 4.10. Workflow for a Practical Application

We demonstrated the actual workflow of MC-SCMNet as shown in Figure 10. In the actual use of pet dog owners and veterinarians, they can take images of dog feces in any environment through cameras or mobile phones, upload them to the cloud, and pass them to the trained MC-SCMNet model for classification. After being processed by this method, they can observe the stool sorting results on computer monitors or mobile phones. The results can help veterinarians diagnose the dog’s intestinal health status, and can also preliminarily judge the dog’s gastrointestinal health status for the owner to improve the dog’s diet in a targeted manner.

## 5. Discussion

In this work, we propose a fine-grained image classification approach for dog feces using MC-SCMNet under complex backgrounds and confirm the effectiveness of MC-SCMNet. In this section, we discuss the use and application value of MC-SCMNet in many aspects and explore how to improve the application of MC-SCMNet in dog feces classification work based on MC-SCMNet and the DFML dataset.

In the process of raising dogs, dogs are easily affected by parasitic infections [6], gastrointestinal biota disorder [7], stress disorders [8], etc., and produce abnormal feces such as diarrhea [12] or hard stool [66]. Different from the manual method of classifying feces by the PURINA FECAL SCORING CHART [19] and the WALTHAM™ Faeces Scoring System [20], novices who cannot distinguish feces can also accurately classify dog feces through MC-SCMNet. In actual use, owners or veterinarians only need to take pictures of dog feces and upload them, and MC-SCMNet can automatically obtain the classification results end-to-end, and provide early warning of the dog’s intestinal health [26]. In the future, we should aim to deploy this approach to edge devices [67] for veterinarians and pet owners.

The results of this study show that the MC-SCMNet model outperforms other classification models in terms of overall classification accuracy under the same experimental setting. Moreover, because we use DSC instead of standard convolution, the total training time of MC-SCMNet is only increased by 16 min and 36 s compared with the original model ResNeSt50 [49]. Although MC-SCMNet has achieved a good balance in time and accuracy compared with ResNeSt, there is still a lot of room for improvement in the time of 3 h 31 min 38 s. Therefore, we will work on developing more lightweight strategies in the future to continuously improve the operational efficiency of the network.

In the performance experiments in Table 6, MC-SCMNet’s accuracy on water-deficient stool (92.06%) is somewhat lower than that of SCMNet (93.65%) and ResNeSt-50 (93.12%). However, it has the maximum accuracy in diarrhea (93.65%), normal (85.71%), and soft stool (84.71%). This shows that MC-SCMNet has no obvious bias in the process of recognizing feces, and has stable classification performance. However, the lack of recognition accuracy also reflects the insufficient amount of data contained in our DFML dataset and the lack of coverage of data categories. To alleviate this situation, we will expand more categories in the future to improve the generalization ability of the model. Since the dataset only contains a small number of samples, this paper uses data augmentation to alleviate the overfitting problem of the network and improve the accuracy of the model, but the improvement effect is limited, not as good as the actual image. Therefore, we plan to extend the DFML dataset in the future to improve model performance.

MC-SCMNet has a good and accurate classification effect on dog feces. However, in practical applications, the results of dog feces classification cannot accurately correspond to certain gastrointestinal diseases. When the dog’s feces have been diagnosed as abnormal by MC-SCMNet for a long time, the owner needs to contact the veterinarian for further diagnosis [26,31]. Therefore, in future work, we should record some characteristics of dogs while taking pictures of dog feces in time, and conduct a certain degree of biochemical analysis on the health of dogs’ gastrointestinal tract without affecting the health of dogs [31]. In this way, the direct correlation between the dog feces and the dog’s gastrointestinal health can be established, and the adequacy and reliability of the work can be improved.

## 6. Conclusions

The dog feces classification model proposed in this paper significantly improves the performance of fine-grained dog feces classification in complex backgrounds. In the absence of a dataset, we built our own DFML dataset for the first time, and proposed MC-SCMNet based on ResNeSt. First, we propose a multi-scale attention down-sampling module, MADM. It expands the information scale and has learnable parameters while focusing on the salient features of feces. This further improves the model’s fecal feature extraction performance. Then, we propose CLAM. It filters the background information interference in the image through bidirectional weight encoding, making the network focus on the target hotspot. Finally, in the feature fusion stage, we combined the characteristics of the above two sub-modules to form the SCM-Block. We use this Block to build a new backbone network SCMNet, which improves the fusion efficiency of the model for dog feces features. Considering the computational cost and training time, we use DSC to reduce the amount of network parameters while maintaining the performance of the model. The experimental results show that MC-SCMNet solves the problems of information loss and false classification in dog feces recognition. Compared to other standard classification models, it achieves the highest accuracy (88.27%) and is more targeted in the fine-grained classification of dog feces. 

In summary, the work in this paper could monitor the health of dogs. The method can provide an initial screening of fecal status to aid veterinary diagnosis when a dog’s feces are abnormal. This can improve the efficiency of the veterinary work while providing an initial warning to the pet dog owner.

## Figures and Tables

**Figure 1 animals-13-01660-f001:**
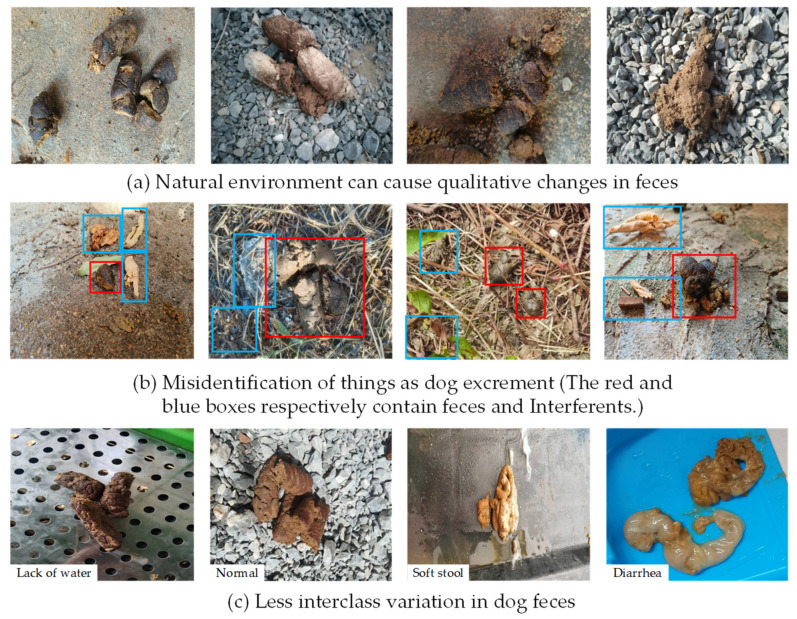
Difficulties in dog feces categorization.

**Figure 2 animals-13-01660-f002:**
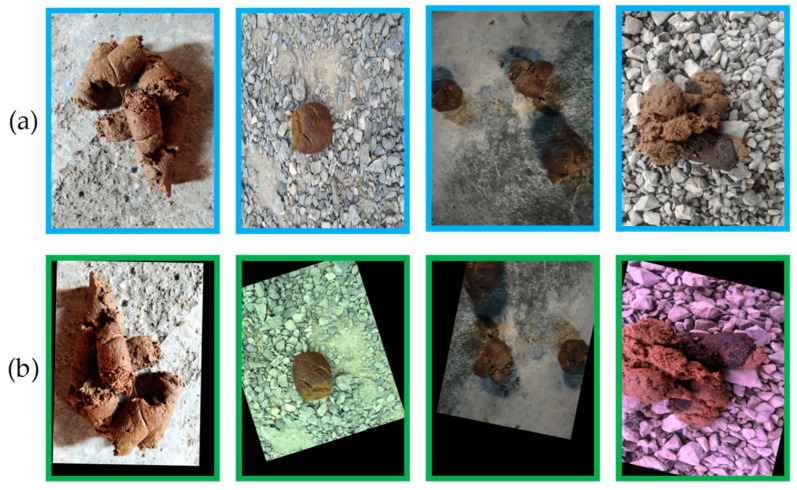
Example images from the DFML dataset: (**a**) Original images, (**b**) images after using data augmentation.

**Figure 3 animals-13-01660-f003:**
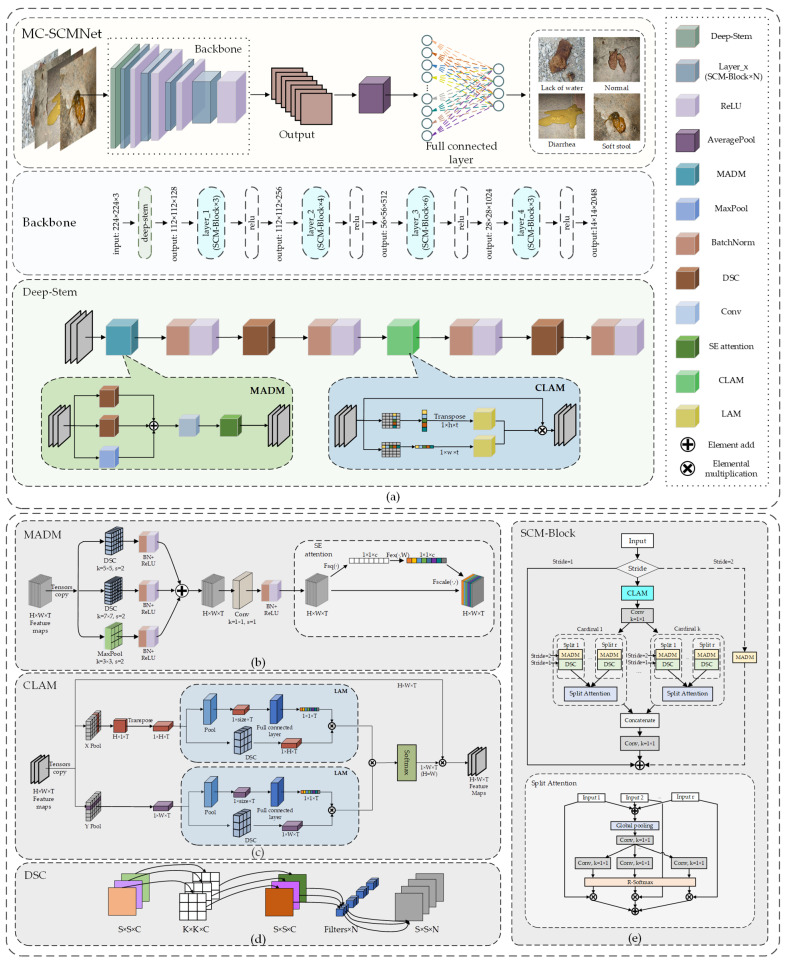
The overall network structure of the proposed MC-SCMNet, with (**a**–**e**) representing the main structure, MADM module, CLAM, DSC, and SCM-Block.

**Figure 4 animals-13-01660-f004:**
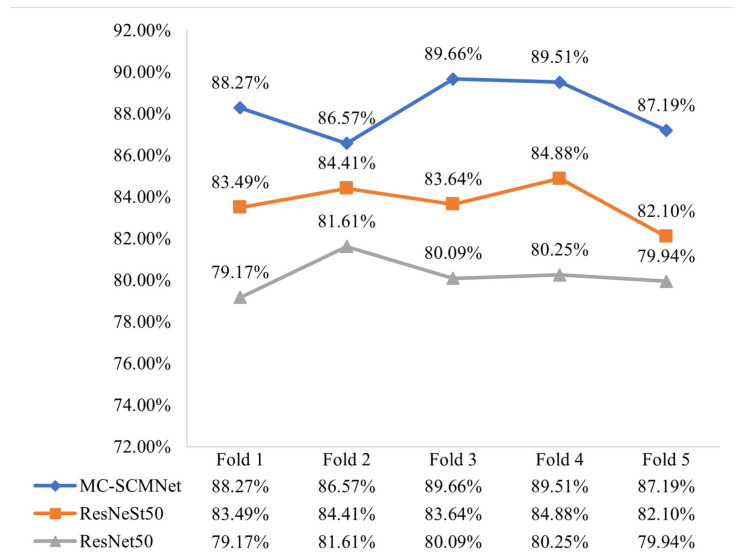
Five-fold cross-validation comparison of MC-SCMNet and ResNet, ResNeSt.

**Figure 5 animals-13-01660-f005:**
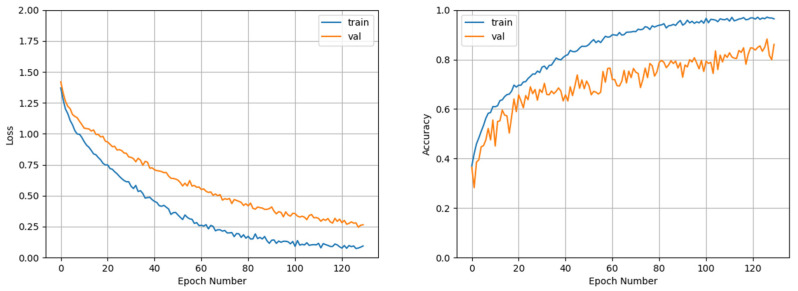
Training loss and accuracy of MC-SCMNet.

**Figure 6 animals-13-01660-f006:**
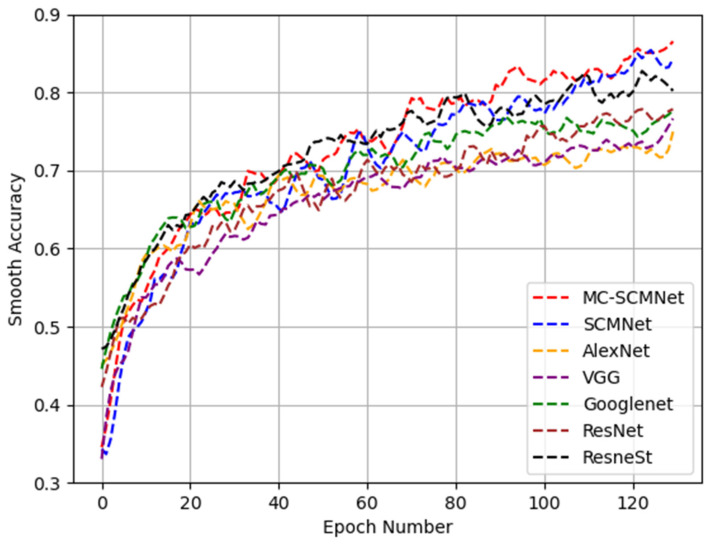
Loss curve of portion various methods.

**Figure 7 animals-13-01660-f007:**
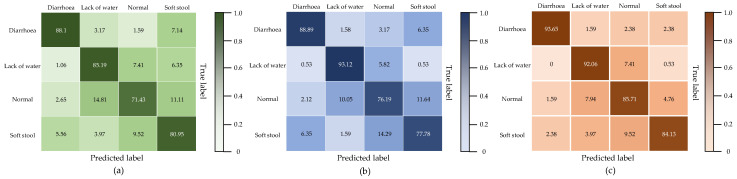
Confusion matrix of the model: (**a**) ResNet50, (**b**) ResNeSt50, (**c**) MC-SCMNet.

**Figure 8 animals-13-01660-f008:**
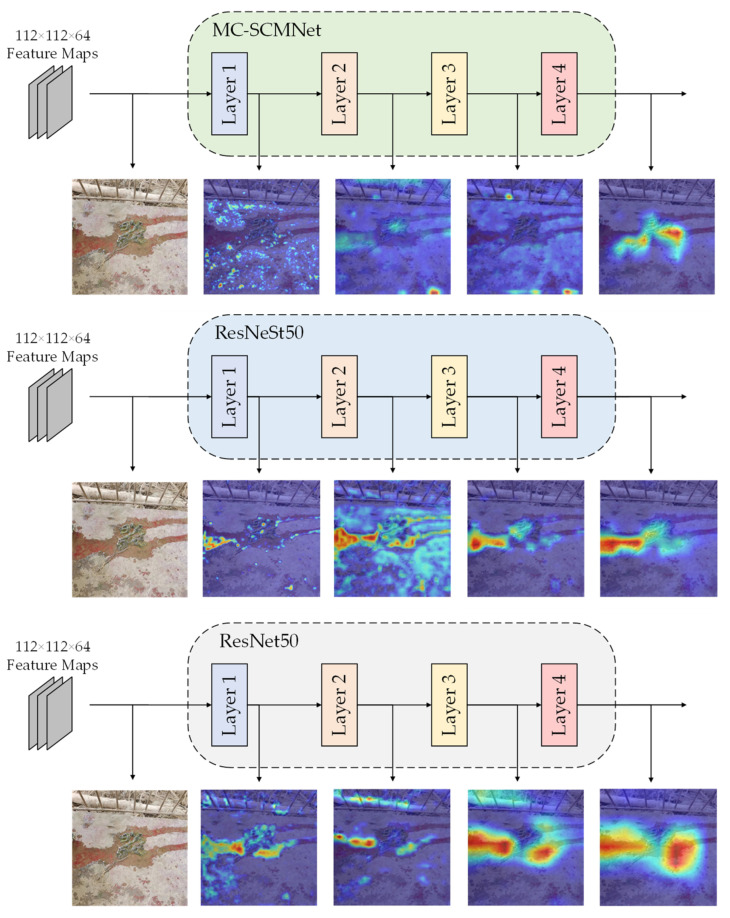
Visual attention results of MC-SCMNet, ResNet50, and ResNeSt50 at each layer.

**Figure 9 animals-13-01660-f009:**
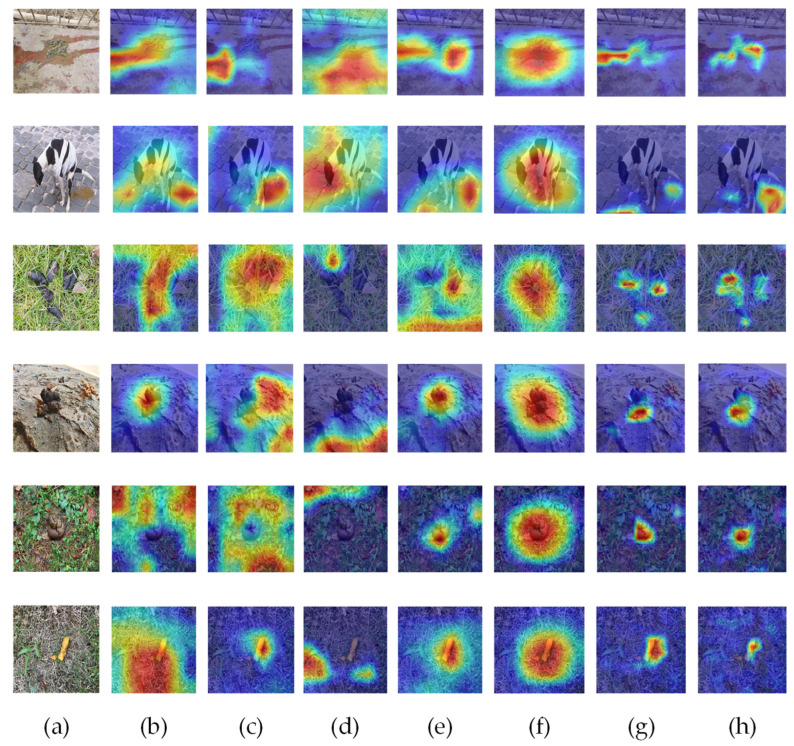
Final visualization results of different models (**a**) Original Images, (**b**) AlexNet, (**c**) VGG-16, (**d**) GoogleNet, (**e**) ResNet-50, (**f**) ResNeXt-50, (**g**) ResNeSt-50, (**h**) MC-SCMNet.

**Figure 10 animals-13-01660-f010:**
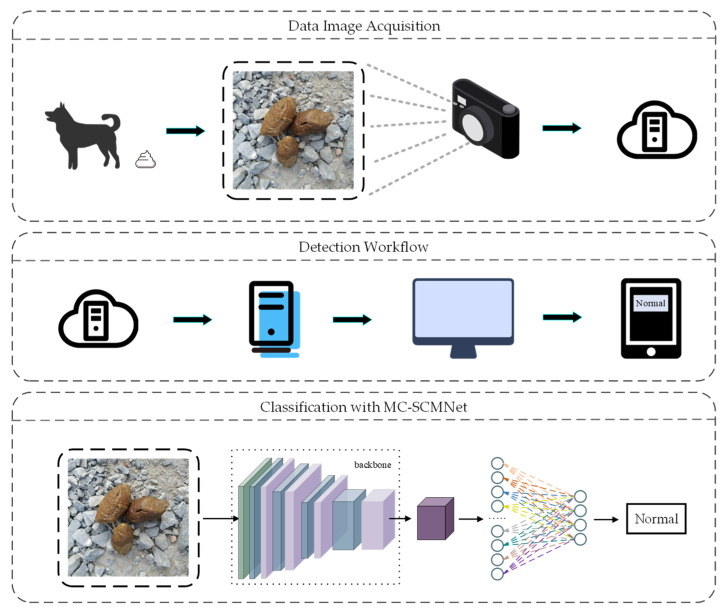
Workflow for a practical application.

**Table 1 animals-13-01660-t001:** The dog feces differentiated by moisture level dataset (DFML).

Category	Example	Characteristics	Number
Lack of water	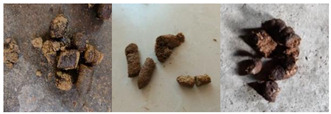	The texture is firm and the surface is dry. Part of the feces is well defined, with a virtually spherical form, and the other part is concave and long sausage shaped.	482
Normal	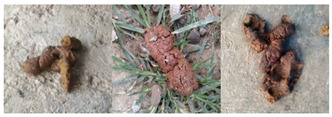	On the surface, there are some obvious fissures that form a serpentine curve.	491
Soft stool	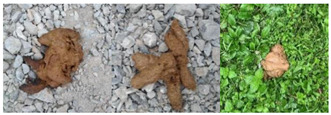	The surface is lustrous and smooth, with a soft touch and irregular shape.	335
Diarrhea	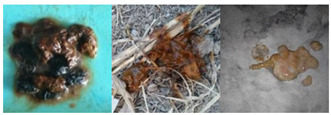	Gooey, highly fluffy, and irregular in shape, partly liquid.	315

**Table 2 animals-13-01660-t002:** Number and percentage of four types of categories following data augmentation.

Category	Number	Proportion
Lack of water	964	29.70%
Normal	982	30.25%
Soft stool	670	20.64%
Diarrhea	630	19.41%

**Table 3 animals-13-01660-t003:** Software and hardware environment settings.

Hardware environment	CPU	12th Gen Intel(R) Core (TM) i9-12900H 2.50 GHz
	GPU	NVIDIA GeForce RTX 3070 Ti
	RAM	16 GB
	Video Memory	8 GB
Software environment	CUDA	11.7
	OS	Windows 11 64-bit operating system
	Pycharm	2022.2.3
	Python	3.7.15
	Torchvision	0.14.0
	Pytorch	1.13.0

**Table 4 animals-13-01660-t004:** Exploring the impact of hyperparameter combinations on MC-SCMNet.

Radix	Cardinal	Accuracy (%)	Params	Training Time
2	1	88.27	33.336 M	3 h 31 min 38 s
2	2	84.10	68.191 M	4 h 11 min 05 s
2	3	80.56	118.612 M	5 h 23 min 54 s
2	4	80.56	184.597 M	6 h 11 min 18 s
3	1	72.53	38.049 M	3 h 58 min 49 s
3	2	84.10	83.230 M	4 h 48 min 22 s
3	3	83.02	149.587 M	6 h 01 min 47 s
4	1	81.17	43.736 M	4 h 08 min 35 s
4	2	78.09	100.903 M	5 h 20 min 19 s

**Table 5 animals-13-01660-t005:** The statistics of different sub-module ablation experiments in MC-SCMNet on the classification accuracy of dog feces.

Schemes	Parameters				Params	Accuracy (%)
	Block	Attention	Conv	Down-Sampling		
Scheme 1	ResBlock	None	Conv	Pooling	25.557 M	80.25
Scheme 2	ResNeSt-Block	Split-Attention	Conv	Pooling	25.579 M	83.64
Scheme 3	ResNeSt-Block	Split-Attention + CA	Conv	Pooling	25.580 M	84.26
Scheme 4	ResNeSt-Block	Split-Attention + CLAM	Conv	Pooling	25.589 M	85.80
Scheme 5	ResNeSt-Block	Split-Attention	DSC	Pooling	16.718 M	83.33
Scheme 6	ResNeSt-Block	Split-Attention + CA	DSC	Pooling	16.719 M	84.10
Scheme 7	ResNeSt-Block	Split-Attention + CLAM	DSC	Pooling	16.728 M	85.96
Scheme 8	SCM-Block	Split-Attention	Conv	Pooling	37.174 M	87.34
Scheme 9	SCM-Block	Split-Attention	Conv	MADM	37.175 M	87.65
Scheme 10	SCM-Block	Split-Attention + CA	Conv	MADM	37.419 M	87.81
Scheme 11	SCM-Block	Split-Attenion + CLAM	Conv	MADM	39.798 M	88.12
Scheme 12	SCM-Block	Split-Attention	DSC	MADM	30.713 M	87.50
Scheme 13	SCM-Block	Split-Attention + CA	DSC	MADM	30.957 M	87.96
Scheme 14	SCM-Block	Split-Attention + CLAM	DSC	MADM	33.336 M	88.27

**Table 6 animals-13-01660-t006:** Accuracy and training time for each basic network method.

Network	Diarrhea	Lack of Water	Normal	Soft Stool	Training Time
AlexNet	65.87%	87.83%	58.73%	43.65%	1 h 48 min 21 s
VGG-16	59.52%	75.13%	76.72%	43.65%	2 h 01 min 18 s
GoogleNet	72.22%	71.95%	75.13%	55.56%	2 h 03 min 11 s
ResNet-50	88.10%	85.19%	71.43%	80.95%	2 h 11 min 55 s
ResNeSt-50	88.89%	93.12%	76.19%	77.78%	3 h 15 min 02 s
SCMNet	89.68%	93.65%	78.84%	80.16%	3 h 22 min 57 s
MC-SCMNet	93.65%	92.06%	85.71%	84.13%	3 h 31 min 38 s

**Table 7 animals-13-01660-t007:** Comparison of MC-SCMNet with ResNet50 and ResNeSt50 evaluation parameters.

Model	Categories	Precision (%)	Recall (%)	*F*_1_ (%)	Accuracy (%)
ResNet50	All categories	80.79	81.16	80.98	80.25
	Diarrhea	88.10	88.80		
	Lack of water	85.19	81.31		
	Normal	71.43	82.82		
	Soft stool	80.95	70.83		
ResNeSt50	All categories	84.13	84.05	84.09	83.64
	Diarrhea	88.89	89.60		
	Lack of water	93.12	88.44		
	Normal	76.19	81.36		
	Soft stool	77.78	75.97		
MC-SCMNet	All categories	88.89	88.93	88.91	88.27
	Diarrhea	93.65	95.16		
	Lack of water	92.06	88.78		
	Normal	85.71	84.82		
	Soft stool	84.13	89.08		

**Table 8 animals-13-01660-t008:** Accuracy and training time for each state-of-the-art method.

Network	Diarrhea	Lack of Water	Normal	Soft Stool	Training Time
ResNeXt50 [48]	87.30%	83.60%	73.54%	79.36%	2 h 40 min 41 s
B-ARNet [28]	90.48%	84.13%	85.19%	80.16%	3 h 23 min 55 s
DMS-Robust Alexnet [62]	88.89%	88.36%	80.95%	83.33%	3 h 42 min 06 s
Swin-transformer [63]	91.27%	90.48%	83.07%	81.75%	3 h 11 min 41 s
CA-MSNet [64]	89.68%	91.53%	82.54%	82.54%	3 h 39 min 32 s
MC-SCMNet	93.65%	92.06%	85.71%	84.13%	3 h 31 min 38 s

**Table 9 animals-13-01660-t009:** One-way ANOVA results.

Source of Variation	F	*p*-Value	F Crit
Between Groups	4.1364	0.000579	2.0666

## Data Availability

Some of the datasets that were used and analyzed in this study have been uploaded to the website https://github.com/ZhouGuoXiong/MC-SCMNet (accessed on 12 March 2023), and all the homemade DFML datasets in this study can be obtained by contacting the corresponding author.

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
