# Peer review of "A Fine-Grained Image Classification Approach for Dog Feces Using MC-SCMNet under Complex Backgrounds"

_animals, 2023, doi:10.3390/ani13101660_

Round 1
Reviewer 1 Report (Previous Reviewer 1)
(1)Please provide the full names of the abbreviations when they first appear in the abstract and text.
E.g. FGVC (Line 84, Page 2)
(2)Please check the manuscript carefully to remove the typos, improve the language and format.
E.g.
-Please check all the reference formats. The sequences of the surnames and give names are different.
...
(3)The length of this paper is too long. Some well-known knowledge and unnecessary experiments can be removed or shortened/condensed, since they can be easily found in textbooks, and are not firstly proposed in this paper. The length of the statements on the contributions at the end of Section 1 is too long.
Author Response
Dear Reviewer,
Thank you very much for your hard work and valuable detailed suggestions on our manuscript. Your suggestion allows us to further improve the content of the paper and also points out the specific direction for our revision. Specifically, we carefully check the language and format of articles to avoid problems such as spelling mistakes. Consider your writing suggestions for the introduction section, we have made appropriate cuts and adjustments in the content to make the paper more compact and smoother. We believe that the revised version allows readers to clearly understand the various parts of the paper. The rest of this letter is our point-by-point response to the suggestions made by you. Comments will be copied, our responses are marked in red, and the content of our manuscript is marked in purple.
Sincerely yours,
The authors.

Reviewer 2 Report (New Reviewer)
This is a very promizing new tool for both canine medicine and canine welfare assessment. Notwithstanding this positive comment, the paper currently raises many formal and methodological questions.
Being based on methods, that are not that well familiar to vets or biologists, such paper especially requires to be very well structured. There is a continuous overlaping or confusion between sections. The Introduction section looks extremely long, but a careful reading shows that it includes data, that should be into the Mat and Meths section (lines 99 to 237). The Mat and Meths section includes many data, that should be presented into the Results section (451 to 593, Tables 4, 5, 6, 7, 8; Figures 4, 5,...). We do find the same confusing presentation into the Discussion section (5.1) with the paragraph commenting Figure 10. All that writing should be in the Results section. The Discussion and Conclusion section, does not clearly discuss or conclude, based on Results and the reader does not really understand what has been achieved with this research.
The Introduction is quita "naive" regarding gastrointestinal disorders in dogs (e.g. nothing said about the role of parasites, or stress-related disorders). The purpose of the research is not clearly stated. Readers expect to obtain some help for the visual evaluation of dogs' feces, but the paper does not clearly show how this technology really helps. The comparizon with other pre-existing methods is not clearly presented (Mat and Mets, Statistics), and does not clearly appear into the Results and Discussion sections.
Regarding the methodology, the authors do not give any information about the health status of the dogs, that produced the feces. Such information would have been extremely interesting to assess the quality of this new methodology.
This enormous amount of work deserves a better writing and a much more structured presentation.
Author Response
Dear Reviewer,
Thank you very much for your hard work and valuable detailed suggestions on our manuscript. Your suggestion allows us to further improve the content and structure of the paper and also points out the specific direction for our revision. We carefully considered and adjusted the structure of the full article to make it easier for readers to understand the work of this paper, especially for veterinarians or biologists. Specifically, we have condensed the introduction and related work to the material on computer vision and restructured the Discussion and Conclusions sections of the article. This has provided a clearer structure to the article, focusing more on the description of feces. We hope that the revised version facilitates veterinarians to quickly understand that the purpose of this article is to assist them in their work by using computer vision to conduct preliminary fecal screening. The rest of this letter is our point-by-point response to the suggestions made by you. Comments will be copied, our responses are marked in red, and the content of our manuscript is marked in purple.
Sincerely yours,
The authors.

Round 2
Reviewer 1 Report (Previous Reviewer 1)
This paper can be accepted now, but the length is too long. It is better that the authors can remove or shorten/condense some well-known knowledge and unnecessary experiments/statements.
Author Response
Dear Reviewer,
Thank you so much for your hard work and recognition of our articles. Considering that there are still some minor English grammar problems and some unnecessary knowledge introductions in the paper, we have made further revisions to the manuscript. We believe that the revised version is more accurate and enables readers to clearly understand the various parts of the paper. The rest of this letter is our point-by-point response to the suggestions made by you. Comments will be copied, our responses are marked in red, and the content of our manuscript is marked in purple.
Sincerely yours,
The authors.

Reviewer 2 Report (New Reviewer)
Thank you for improving the structure of the paper and for making th reading more comfortable.
Author Response
Dear Reviewer,
Thank you so much for your hard work and recognition of our articles. Considering that there are still some minor English grammar problems and some unnecessary knowledge introductions in the paper, we have made further revisions to the manuscript. We believe that the revised version is more accurate and enables readers to clearly understand the various parts of the paper.
Sincerely yours,
The authors.

This manuscript is a resubmission of an earlier submission. The following is a list of the peer review reports and author responses from that submission.
Round 1
Reviewer 1 Report
(1)Why is MADS (not MADM) short for “multi-scale attention down-sampling module"
(2)The contributions at the end of Section 1 are exaggerated. The authors used some steps and technologies that are firstly proposed by them. “Propose” is not appropriate. “Design”, “Develop”, “Improve”, “Combine”, “Employ” “Modify” are better.
(3)The length of this paper is too long. Some well-known knowledge can be removed or shortened/condensed, since they can be easily found in textbooks, and are not firstly proposed in this paper.
(4)The review of the related works and comparison experiments can be more sufficient. Please carefully read, cite and compare (if applicable) the following papers.
Feces recognition: -E-health self-help diagnosis from feces images in real scenes; -A light-weight practical framework for feces detection and trait recognition
Multi-scale fusion in deep learning: -Mask Refined R-CNN: A network for refining object details in instance segmentation; -Object detection based on multi-layer convolution feature fusion and online hard example mining
If the authors cannot employ these methods or compare their method with these methods, at least they could introduce/mention these novel technologies in related sections to improve the quality of the survey.
(5)Please provide and label the reference indices of the compared methods in the figures and tables, and then the readers can judge whether the compared methods are SOTA.
(6)Please use bold font to label the best results in all tables.
Author Response
Dear Reviewer,
Thank you very much for your hard work and valuable detailed suggestions on our manuscript. Your suggestion allows us to further improve the content of the paper, and also points out the specific direction for our revision. Specifically, we restructured the paper and created a new subpoint "Related Work". Considering the relevance of the two parts, we have made appropriate cuts and adjustments in the content to make the paper more compact and smoother. We believe that the revised version allows readers to clearly understand the various parts of the paper.
Sincerely yours,
The authors.
Reviewer 2 Report
This manuscript is about a fine-grained image classification approach for dog feces using MC-SCMNet under complex backgrounds.
The manuscript is well written and presented.
However, I have some remarks:
Line 43-44 add reference
Line 131-132 : delate the sentence is repeated twice
Line 137: add reference to the figure 1
Line 165 : arrange space between words
I have some questions about the method
This methods considered as a complementary diagnosis, the veterinarian could identify if feces are normal or no, without using any kind of program.
Is this method effective and sufficient to make a good diagnosis, without the presence of veterinarian ? Other diseases such as parasitosis, hepatitis ......... could be detected with this method ?
Author Response
Dear Reviewer,
Thank you very much for your recognition of our work. Your suggestion allows us to further improve the content of the paper, and also points out the specific direction for our revision. Specifically, we restructured the paper and created a new subpoint "Related Work". Considering the relevance of the two parts, we have made appropriate cuts and adjustments in the content to make the paper more compact and smoother. We believe that the revised version allows readers to clearly understand the various parts of the paper.
Sincerely yours,
The authors.

Reviewer 3 Report
Line 15: ‘dog raising industry’??? – I beg your pardon, WHAT industry?
Lines 16-17: ‘We can prevent gastrointestinal illnesses in dogs by monitoring their feces.’ – NO, we cannot. By observation we can try to diagnose the culprit of the change in consistency/volume and act accordingly.
Lines 17-18: ‘Nowadays, fecal status(???) is detected(???) mostly by veterinarians (NOT TRUE), however this method is inefficient’ – NO, it is well accepted and commonly adapted, including numerous scientific publications. Suggested reference: Waltham fecal scoring booklet (https://www.waltham.com/resources/waltham-booklets) and for example: Vendramini et al. Evaluation of the influence of coprophagic behavior on the digestibility of dietary nutrients and fecal fermentation products in adult dogs. Vet. Sci. 2022, 9, 686. https://doi.org/10.3390/vetsci9120686.
Lines 19-20: ‘...gathers images of dog excrement in natural environments...’ – do you mean that the value of this manuscript is to mention gathering many feces photographs (from different sources)?
Suggested reference: Waltham fecal scoring booklet (https://www.waltham.com/resources/waltham-booklets).
Lines 42-45: ‘Dog rearing provides two economic benefits: food and pets. As food, dog meat is tasty and nutritional; as pets, they are gaining popularity. Therefore, the economic benefits of feeding dogs in large amounts on farms are higher.’ - could you explain your shocking declarations on “dog rearing”, namely, what are you referring to? What dog farms are you mentioning, suggesting/supporting increasing economic benefits of this “industry”? On what grounds you claim the “tastiness” of dog meat?
Line 53: define ‘feces identification’. Once again, in experimental setups or in home-lead experiments stool COLLECTION and analysis is well known and described with NO NEED for any kind of IDENTIFICATION. Suggested reference: Waltham fecal scoring booklet (https://www.waltham.com/resources/waltham-booklets).
Lines 57-58: ‘This process takes a long time, and the amount of detection toughly satisfy large-scale breeding’ – what do you mean by ‘satisfy large scale breeding’?
Lines 59-60: ‘At the same time, people's subjective view will influence detection quality.’ – define ‘detection quality’. And: there are (likely) no available scientific reports noting strongly biased results of stool quality evaluation.
Lines 66-67: ‘This paper will improve further the image recognition technology based on deep learning for the identification of dog feces.’ – such claim is simply UNSUBSTANTIATED.
Lines 139-140: ‘...in specifically...’ – what does it mean?
Lines 184-185: ‘According to expert recommendations’ – what experts? what recommendations? Suggested reference: Waltham fecal scoring booklet (https://www.waltham.com/resources/waltham-booklets).
Lines 216-218: ‘The following is the rest of the paper...’ - needs to be DELETED.
Line 223: ‘On the advice of specialists...’ - what advice? what specialists?
Author Response
Dear Reviewer,
Thank you very much for your hard work and valuable detailed suggestions on our manuscript. Your suggestion allowed us to further improve the content of the paper, and also made us realize that there are language problems in many aspects. With the help of Elsevier Language Editing Service, we checked and corrected the full text sentence by sentence to reduce spelling errors as much as possible and improve the fluency of reading. In terms of content, in consideration of cultural differences, we only elaborate on the raising of pet dogs, and delete the part about raising dogs as food. Structurally, we have readjusted and created a new subpoint "Related Work". Considering the relevance of the two parts, we have made appropriate cuts and adjustments in the content to make the paper more compact and smoother. We believe that the revised version will give readers a clear understanding of the various parts of the paper and a better understanding of the work in this paper.
Sincerely yours,
The authors.

Reviewer 4 Report
The manuscript bening in the category of ,,Article'' and not a ,,Review'' I consider that the ,,Introduction'' is too large. Please try to short it at maximum 1-1.5 pages length.
The findings of other authors can be moved to Discussions.
It is a very interesting study, but I consider it is a pilot study. Are missing the comparissons with the standard chemical and medical analysis. To appreciat the dog's health status by the stool, only by some photos and statistical measurements, it's not enought if there is no medical evaluation.
The study, to be real, the photos of feces have to be compared with some chemical analysis of the feces, some biochemical analysis of dog's blood and some medical evaluation of dogs.
This manuscript present in details the techniques they used, but are missing the comparisons with chemical and medical evaluation of the feces and of the dogs.
Please present the ,,Conclusions'' separately by the discussions
Author Response
Dear Reviewer,
Thank you very much for your hard work and valuable detailed suggestions on our manuscript. And thank you for your recognition of our work. Your suggestion allows us to further improve the content of the article, and also points out the specific direction for our revision. We have restructured the article and created a new subpoint "Related work". Considering the relevance of the two parts, we have made appropriate cuts and adjustments in the content to make the paper more compact and smoother. We believe that the revised version allows readers to clearly understand the various parts of the paper.
Sincerely yours,
The authors.
